# Peer Relationships and Psychosocial Difficulties in Adolescents: Evidence from a Clinical Pediatric Sample

**DOI:** 10.3390/jcm14207177

**Published:** 2025-10-11

**Authors:** Leonardo Tadonio, Antonella Giudice, Claudia Infantino, Simone Pilloni, Matteo Verdesca, Viviana Patianna, Gilberto Gerra, Susanna Esposito

**Affiliations:** 1Department of Mental Health, AUSL Parma, 43126 Parma, Italy; ltadonio@ausl.pr.it (L.T.); ggerra@ausl.pr.it (G.G.); 2Pediatric Clinic, Department of Medicine and Surgery, University of Parma, 43126 Parma, Italy; antonella.giudice@unipr.it (A.G.); infantinoclaudia@gmail.com (C.I.); simone.pilloni@unipr.it (S.P.); matteo.verdesca@unipr.it (M.V.); vpatianna@ao.pr.it (V.P.)

**Keywords:** adolescent mental health, friendship satisfaction, prosocial behavior, peer relationships, eating-disorder risk, substance use

## Abstract

**Background**: Adolescence is a critical developmental stage marked by vulnerability to psychological difficulties. While family relationships, peer bonds, prosocial behaviors, and health-risk factors have been linked to adolescent mental health, few studies have examined their joint effects in clinical pediatric populations. This study assessed demographic, clinical, relational, and behavioral predictors of psychological difficulties in Italian adolescents. **Methods**: A cross-sectional sample of 177 adolescents (aged 11–14 years) from a pediatric clinic completed the Strengths and Difficulties Questionnaire (SDQ). The Total Difficulties (SDQ TD) score was the main outcome. Associations were tested with ordinary least squares (OLS) and confirmed using robust MM regression. Bootstrap confidence intervals and Benjamini–Hochberg corrections were applied. Sensitivity analyses excluded the Peer Problems subscale to address part–whole overlap. **Results**: Higher friendship satisfaction was consistently associated with fewer psychological difficulties, confirming its role as a strong protective factor. Prosocial behavior and male sex were also linked to fewer difficulties in initial analyses, though these associations were less stable after correction. Sensitivity analyses further supported the protective value of friendship satisfaction, even when accounting for overlap with peer problems. Despite relatively low overall levels of psychological difficulties, nearly one-quarter of adolescents met the clinical cut-off for eating disorder risk. **Conclusions**: Friendship satisfaction was the strongest protective factor, while prosocial behavior and sex showed weaker consistency. Findings suggest that distinct aspects of peer relationships jointly shape adolescents’ psychological outcomes. Interventions promoting social functioning may support mental health in clinical youth populations.

## 1. Introduction

Adolescence is a key developmental stage, usually covering the second decade of life, marked by major biological, psychological, and social changes. These include brain maturation, identity formation, greater independence from parents, and stronger focus on peers [1,2,3]. While this period offers growth opportunities, it also brings higher vulnerability to emotional and behavioral problems [4,5], with long-term effects on adult mental health [6].

The Strengths and Difficulties Questionnaire (SDQ) is widely used to assess adolescent psychosocial functioning [7]. It measures both difficulties (emotional, conduct, hyperactivity–inattention, peer problems) and strengths (prosocial behavior). High SDQ Total Difficulties (TD) scores are linked to poor school outcomes, psychiatric diagnoses, and social exclusion [8,9]. These difficulties are influenced by multiple demographic, medical, relational, and behavioral factors.

Family relationships are central to adolescent well-being. Warm, supportive parent-child bonds predict fewer emotional and behavioral problems [10,11,12] by strengthening resilience, self-esteem, and emotional regulation [13,14]. As peer time increases, friendships become equally important [15,16]. Satisfying friendships protect against loneliness, depression, and behavioral issues [17,18,19], while poor friendships or conflict increase distress [20].

Prosocial behavior—helping, sharing, comforting—has also been tied to better emotional adjustment and stronger social skills [21,22]. Adolescents who act prosocially often report higher peer acceptance, self-worth, and fewer behavioral problems [23].

Physical health is another factor. Chronic conditions (e.g., asthma, endocrine or gastrointestinal disorders) can add stress, limit activities, and raise concerns about health or body image [24,25]. Such conditions are linked to higher depressive symptoms and lower quality of life [26].

Risk behaviors complicate this picture. Alcohol and nicotine use, even at low levels, are associated with psychological difficulties [27,28,29,30]. Cannabis use has been linked to cognitive and educational problems and greater psychiatric risk [31]. Substance-use habits formed in adolescence often persist into adulthood, highlighting the need for early detection [32].

Despite rich evidence, few studies have explored these factors together in clinical pediatric populations. Adolescents in pediatric clinics face both medical and developmental challenges [33]. Cultural context also matters: in Italy, family ties are emphasized, but peers and autonomy are increasingly important, sometimes creating tensions [34,35,36].

Compared to community samples, clinical adolescents may deal with added stress from chronic illness, medical monitoring, and disrupted routines, which can affect social life and increase psychosocial vulnerability [37]. Studying them is crucial for understanding both general and context-specific influences on adjustment and for guiding interventions. Pediatric clinics are also well-placed for early detection of emotional and behavioral issues.

We developed four hypotheses:H1: Higher friendship satisfaction predicts fewer difficulties.H2: Greater prosocial behavior predicts fewer difficulties.H3: Females report more difficulties than males.H4: Chronic illness, eating disorder risk, and substance use predict more difficulties.

This study tests how demographic, relational, behavioral, and clinical factors jointly relate to psychological difficulties in Italian adolescents attending a pediatric clinic. Identifying key predictors in this setting can inform psychosocial screening and support in pediatric care.

## 2. Methods

### 2.1. Study Design and Population

This cross-sectional observational study was conducted in a pediatric outpatient clinic in Italy (Parma University Hospital, Parma, Italy) and involved a consecutive sample of 177 adolescents aged between 11 and 14 years. Recruitment took place during routine clinical appointments, and both adolescents and their parents or legal guardians were invited to participate. Eligibility criteria included being within the specified age range and demonstrating sufficient comprehension to complete a set of self-report questionnaires in Italian. Adolescents were excluded if they presented with acute medical emergencies, severe psychiatric crises requiring immediate intervention, or cognitive impairments that would prevent reliable self-reporting. Prior to participation, written informed consent was obtained from the parents or guardians, and assent was secured from the adolescent. The self-report questionnaires were administered anonymously and without the presence of parents or guardians to ensure privacy and minimize potential response biases, particularly social desirability and parental influence [38]. The study protocol was approved by the relevant institutional ethics committee and adhered to the principles outlined in the Declaration of Helsinki.

### 2.2. Measures

Psychological difficulties were assessed using the Italian self-report form of the Strengths and Difficulties Questionnaire (SDQ) [7] for adolescents aged 11–17 years, focusing on the Total Difficulties (SDQ TD) score. This SDQ TD was derived from the sum of the Emotional Symptoms, Conduct Problems, Hyperactivity-nattention, and Peer Problems subscales (Cronbach’s α = 0.78 in the present sample). It ranges from 0 to 40, with higher values indicating greater psychological difficulties.

Medical conditions were clinician-assessed and classified as acute or chronic. All other predictors were obtained via adolescent self-report. Demographic variables included age, sex, and citizenship (Italian vs. non-Italian). Relational variables comprised mother-relationship, father-relationship, and friendship satisfaction, each assessed with a single-item Likert-type rating from 1 (not at all satisfied) to 5 (very satisfied). Eating disorder (ED) risk was assessed with the SCOFF [39] and coded as positive when the score was ≥3, following validation in Italian samples [40]. Prosocial behavior was measured using the SDQ Prosocial subscale (Cronbach’s α = 0.72), with higher scores reflecting greater prosocial tendencies. Adolescents indicated whether they had used alcohol, nicotine, or cannabis in their lifetime, during the past year, and in the past-30-day, with responses coded as yes/no. Nicotine use encompassed both traditional tobacco smoking and the use of electronic nicotine delivery systems, such as e-cigarettes.

Although normative data for the Italian self-report form of the SDQ are currently lacking, this version of the instrument has been previously used in Italian adolescent populations [41,42]. While the three- and five-factor models of the SDQ are frequently used in the literature, their structure has not been consistently replicated across cultures [43]. Based on this, we used the SDQ Total Difficulties score and the Prosocial subscale, which represent the two core dimensions of the instrument.

The SCOFF questionnaire has previously been used as a screening tool for eating disorder risk in adolescent studies, both in community samples and in clinical populations, such as adolescents with type 1 diabetes [44,45].

### 2.3. Statistical Analysis

All analyses were conducted on the complete dataset, as no data were missing. The primary outcome was the SDQ Total Difficulties (SDQ TD) score (Cronbach’s α = 0.78), which showed an approximately symmetric distribution (skewness = 0.30; excess kurtosis = −0.58) without ceiling or floor effects. Although the Shapiro-Wilk test indicated non-normality (*p* = 0.006), inspection of the histogram and Q-Q plot suggested only minor departures, supporting its treatment as a continuous outcome. Bivariate associations among predictors were examined using Spearman’s *ρ* and rank-biserial correlations (for binary-continuous pairs). To reduce redundancy, only past-30-day substance-use indicators were retained; cannabis use was excluded due to its very low prevalence (2/177).

We fitted an ordinary least-squares (OLS) multiple regression including 11 prespecified predictors (five continuous, six binary; all binary variables dummy-coded). Model diagnostics supported OLS assumptions: scatterplots and component-plus-residual plots indicated linearity of continuous predictors, variance inflation factors were <2, and homoscedasticity and residual normality were supported by both formal tests and visual inspection. Influential cases were identified Via leverage, Cook’s distance, DFFITS, and DFBETAS [46]. Given their presence, we also fitted a robust MM regression [47,48] as a secondary confirmatory analysis, using the robust-base package (Tukey’s bisquare ψ-function, default tuning c = 4.685, ~95% efficiency under normality). This estimator down-weights the influence of outliers and high-leverage points while preserving the interpretability of coefficients. For each model, both unstandardized (B) and standardized (*β*) coefficients were estimated, with the latter obtained by z-standardizing the outcome and predictors prior to model fitting.

Uncertainty for both OLS and robust models was quantified Via nonparametric pairs (case) bootstrap (boot package; R = 5000; seed = 1234), resampling rows with replacement and refitting the model at each resample. For unstandardized coefficients, we report 95% bias-corrected and accelerated (BCa) confidence intervals (CI) and bootstrap standard errors [49]. Two-sided bootstrap *p*-values were adjusted for multiplicity via the Benjamini-Hochberg (BH) procedure (*q* = 0.05) for false discovery rate (FDR) control within each model [50].

This analytical strategy was adopted to strengthen the robustness of findings based on real-world clinical data. We used robust MM regression to confirm the OLS findings, assess the stability of results, and reduce the impact of outliers. The bootstrap procedures supported inference under non-normality and in small samples, while FDR correction offers a balanced trade-off between sensitivity (true positives) and specificity (false discovery control).

A targeted sensitivity analysis was conducted to address potential part-whole overlap between the Peer Relationships Problems subscale from the SDQ TD and the predictor Friendship Satisfaction. Both OLS and robust MM models were re-estimated using SDQ TD15—SDQ TD, excluding the Peer Problems subscale, under the same bootstrap and BH adjustment procedures.

All analyses were performed in R version 4.3.2 (R Core Team, 2023) [51].

## 3. Results

The sample consists of 177 adolescents aged 11–14 years (M = 12.51, SD = 1.11). Males represented 56.5% (*n* = 100) of the participants, while 93.2% (*n* = 165) were Italian citizens. Clinicians assessed that 71.8% (*n* = 127) of the sample had a chronic medical condition. Relationship satisfaction was high with both parents (mothers: M = 4.37, SD = 0.83; fathers: M = 4.21, SD = 0.96) as well as with friends (M = 4.22, SD = 0.88). The mean SDQ TD score was 10.62 (SD = 5.46), and the mean prosocial behavior score was 7.62 (SD = 1.99). As a whole, these results suggest low levels of psychological difficulties and normative prosocial functioning. However, 23.2% (*n* = 41) of adolescents were classified as at risk for eating disorders according to the SCOFF clinical cut-off. Overall, rates of substance use were relatively low across all substances and time frames assessed. An exception was lifetime alcohol use reported by 19.8% (*n* = 35) of participants. In the past 30 days, alcohol, nicotine, and cannabis use were disclosed by 9.0% (*n* = 16), 7.3% (*n* = 13), and 1.1% (*n* = 2) of adolescents, respectively. Descriptive statistics are presented in Table 1.

Spearman’s and rank-biserial correlation coefficients are reported in Appendix A. Mother-relationship and father–relationship satisfaction were strongly intercorrelated (*ρ* = 0.69), whereas friendship satisfaction showed only weak, non-significant associations with mother–adolescent (*ρ* = 0.15) and father-adolescent satisfaction (*ρ* = 0.12), consistent with a distinct construct. Prosocial behavior correlated positively with mother-adolescent (*ρ* = 0.30) and father-adolescent satisfaction (*ρ* = 0.32). Age was moderately negatively associated with parental satisfaction (*ρ* = −0.24 to −0.31). Indicators of alcohol and nicotine use across all time frames were strongly intercorrelated (*ρ* = 0.63–0.88); therefore, only past-30-day indicators were included in the regression analyses. Cannabis (past 30 days) was excluded because of very low prevalence (2/177). Eating disorder risk showed small positive associations with alcohol and nicotine use (*ρ* = 0.17–0.25) but no significant associations with parental or friendship satisfaction.

We then tested the simultaneous effects of all predictors on SDQ TD using an OLS multiple linear regression (Table 2).

The model explained 41% of the variance (R^2^ = 0.41, adj. R^2^ = 0.37; *F*(11, 165) = 10.46, *p* < 0.001). After BH adjustment of bootstrap *p*-values, three predictors remained statistically significant: friendship satisfaction (B = −2.05, 95% BCa CI [−2.71, −1.31], *p*_BH_ < 0.001), sex (female = 1; B = 2.30, 95% BCa CI [0.91, 3.70], *p*_BH_ = 0.002), and prosocial behavior (B = −0.64, 95% BCa CI [−1.04, −0.27], *p*_BH_ = 0.006). Father-relationship satisfaction showed an inverse association with psychological difficulties, but did not retain significance after FDR adjustment (*p*_BH_ = 0.054). All other variables—age, chronic condition, Italian citizenship, mother-relationship satisfaction, ED risk, and past-30-day alcohol and nicotine use—were not statistically significant after adjustment. Comparing the relative magnitudes of the standardized coefficients, friendship satisfaction (β = −0.33) emerged as the strongest protective factor. Prosocial behavior (*β* = −0.23, *p*_BH_ = 0.006) and female sex (*β* = 0.21, *p*_BH_ = 0.002) were also significantly associated with SDQ TD, indicating respectively a protective effect and greater difficulties among adolescent girls.

As a targeted sensitivity analysis to address potential part-whole overlap between friendship satisfaction and the Peer Problems subscale, we re-estimated the models using SDQ TD15 (excluding Peer Problems). In these models, the strength of the friendship satisfaction association was attenuated, with the standardized coefficient decreasing from *β*_OLS_ = −0.33 (SDQ TD) to *β*_OLS_ = −0.24 (SDQ TD15) and from *β*_robust_ = −0.34 to −0.23, corresponding to a ≈27–32% reduction in magnitude. In the OLS model, the effect remained statistically significant after BH adjustment (*p*_BH_ < 0.001), whereas in the robust MM model, it was of similar magnitude but did not reach the conventional significance threshold (*p*_BH_ = 0.101). In both models, however, the unadjusted 95% BCa CIs excluded zero (Appendix A).

As shown in Table 2 (last column), we assessed the sensitivity of coefficients to influential observations, calculated as Δ = B_robust_ − B_OLS_. Coefficients with |Δ| ≤ 0.10 were considered stable. Friendship satisfaction (|Δ| = 0.04), prosocial behavior (0.07), ED risk (0.06), and past-30-day alcohol use (0.01) met this criterion. Father-relationship satisfaction was borderline (|Δ| = 0.11). All other variables—age (0.28), chronic condition (0.33), Italian citizenship (0.17), mother–relationship satisfaction (0.20), and past-30-day nicotine use (0.73)—exceeded this threshold, indicating greater susceptibility to influential cases.

To facilitate direct comparison of effect sizes and statistical significance across estimation methods, we present a forest plot in Figure 1.

Controlling for all covariates, friendship satisfaction showed the largest association with SDQ TD: higher satisfaction was linked to fewer difficulties, and this effect was robust across both OLS and robust MM estimators. It was the only predictor to retain statistical significance after BH adjustment in both models, underscoring its stable protective role. In the primary OLS analysis, prosocial behavior and sex also remained significant after BH correction. However, although both estimators yielded 95% BCa CIs excluding zero, these two effects did not survive BH adjustment in the robust MM model, indicating some sensitivity to the down-weighting of influential observations.

The Appendix A provide: correlation matrix among study variables (*p*-values BH-adjusted) (Appendix A); the correlation matrix of the study variables (Appendix A); OLS model influence diagnostics (SDQ TD) (Appendix A); variance inflation factors (VIF) for predictors in the OLS model (SDQ TD) (Appendix A); comparison of OLS and robust MM regression estimates for SDQ TD outcome; (Appendix A) comparison of OLS and robust MM regression estimates for SDQ TD15 outcome (Appendix A); model fit statistics for OLS and robust MM regression models with SDQ TD and SDQ TD15 outcomes (Appendix A); distribution of SDQ Total Difficulties score (SDQ TD) (Appendix A); component−plus−residual (partial) plots for continuous predictors in the OLS model with SDQ TD as the outcome (Appendix A); standard diag-nostic plots for the OLS regression model with SDQ TD as the outcome: residuals vs. fitted values, scale–location plot, Q–Q plot of residuals, and residuals vs. leverage (Appendix A);. influence diag-nostics thresholds and counts for OLS model (Appendix A).

## 4. Discussion

This study examined how demographic, medical, relational, behavioral, and prosocial factors relate to psychological difficulties in Italian adolescents attending a pediatric clinic. Using OLS and robust MM regression with bootstrap and FDR correction, we aimed to identify stable predictors. Our results add to existing research on adolescent mental health. Consistent with prior studies, friendship satisfaction emerged as the strongest protective factor [16,18,21,52]. Unlike earlier findings, however, parental satisfaction and substance use were not significant, possibly due to high parental satisfaction levels and the low prevalence of substance use in our sample [10,11,12,25,26,27].

Friendship satisfaction showed the largest and most consistent effect, supporting evidence that positive peer bonds buffer against stress and reduce symptoms [20,53]. Its stability across methods, even after excluding peer problems from the SDQ, suggests a robust protective role. Cultural factors may influence this effect: in Italy, where family ties are emphasized but peer influence is growing, friendships may take on particular importance [33,34,35]. For adolescents with chronic illness, friendships may provide vital support when other social opportunities are limited [24].

Prosocial behavior was linked to fewer difficulties in OLS models, consistent with theories that helping others fosters peer acceptance, empathy, and self-worth [23,24,54,55,56,57]. However, this effect was weaker after correction, suggesting less robust evidence. Female sex was associated with greater difficulties, aligning with epidemiological studies showing higher internalizing symptoms among girls [58,59,60,61,62], though this effect also weakened in robust analyses.

Contrary to some studies [13,63,64], parental satisfaction, chronic medical conditions, eating disorder (ED) risk, and substance use were not significant predictors. For chronic conditions, psychosocial outcomes may depend more on perceived illness impact than diagnosis alone [26,27]. ED risk was also unrelated, perhaps because early eating concerns reflect a mix of personal and environmental factors [65,66], and the SDQ may be less sensitive to ED-specific problems [67]. Low prevalence and binary assessment of substance use likely reduced power to detect associations.

Clinically, the findings highlight the importance of assessing friendship quality. Interventions that promote positive peer interactions could protect adolescent mental health in both schools and clinics [65,66,67,68]. Prosocial behavior may also serve as a modifiable factor, though evidence was less consistent [69,70]. Gender-sensitive approaches may help address girls’ heightened vulnerabilities.

Strengths of this study include a well-defined clinical sample, simultaneous testing of multiple predictors, and robust statistical methods. Limitations include the modest sample size, cross-sectional design, single-item measures for relationships, and omission of key confounders such as SES and family structure. Importantly, we did not assess medication use or BMI, both of which may influence psychological functioning and should be considered in future work. Finally, the clinical context improves relevance for pediatric care but may reduce comparability with community samples.

Future research should use larger and longitudinal samples, explore cultural and illness-specific contexts, and test interventions targeting friendship quality and prosocial behavior. As online and offline peer interactions increasingly shape adolescent development, their combined effects on mental health warrant investigation [71].

## 5. Conclusions

In this clinical sample of Italian adolescents, friendship satisfaction emerged as the most stable protective factor against psychological difficulties. Prosocial behavior and sex were also linked to fewer difficulties, although these associations did not remain significant in confirmatory analyses. Alongside friendship satisfaction, low levels of peer problems appeared to be important, as shown by the attenuation of the effect once the Peer Problems subscale was excluded. Taken together, these findings show that peer relationships play a multidimensional role in adolescent mental health—through both friendship satisfaction and difficulties in peer relationships—and underscore the importance of pediatric care interventions that support adolescents’ social functioning and the development of supportive peer experiences.

## Figures and Tables

**Figure 1 jcm-14-07177-f001:**
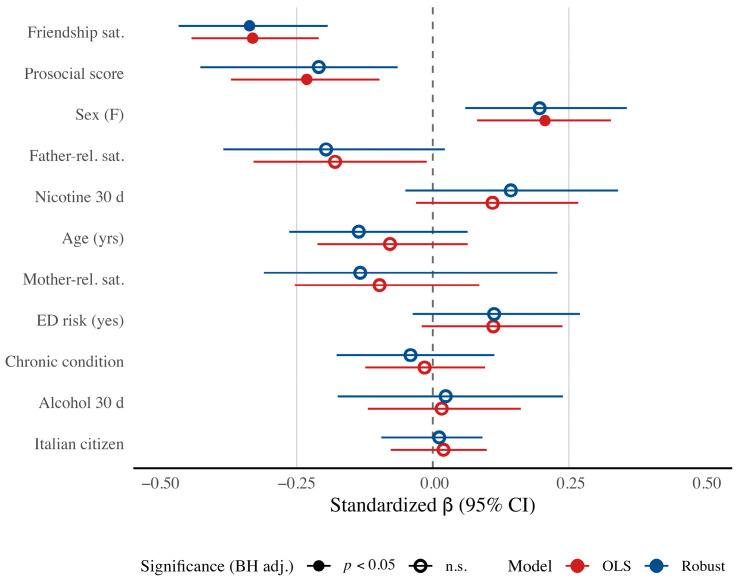
Standardized coefficients (*β*) with 95% BCa CI from OLS and robust MM models predicting SDQ Total Difficulties (SDQ TD). Note. Standardized coefficients (*β*) with 95% BCa confidence intervals (bootstrap, R = 5000) from OLS (red) and robust MM (blue) regression models predicting SDQ Total Difficulties. Filled symbols indicate BH-adjusted *p* < 0.05; open symbols indicate non-significant predictors. Reference lines shown at *β* = 0 (dashed) and ±0.25 (solid). Negative values indicate protective effects; positive values indicate risk.

**Table 1 jcm-14-07177-t001:** Descriptive statistics for adolescents attending a pediatric clinic (*N* = 177).

Variable	M	SD
Continuous variables		
Age (years)	12.51	1.11
SDQ Total Difficulties	10.62	5.46
Prosocial behavior (SDQ)	7.62	1.99
Mother-relationship satisfaction	4.37	0.83
Father-relationship satisfaction	4.21	0.96
Friendship satisfaction	4.22	0.88
Categorical variables	*n*	%
Sex (Female)	77	43.5
Italian citizenship (Yes)	165	93.2
Chronic condition (Yes)	127	71.8
Eating-disorder risk (SCOFF ≥ 3)	41	23.2
Alcohol use, lifetime (Yes)	35	19.8
Alcohol use, past year (Yes)	24	13.6
Alcohol use, past 30 days (Yes)	16	9.0
Nicotine use, lifetime (Yes)	18	10.2
Nicotine use, past year (Yes)	14	7.9
Nicotine use, past 30 days (Yes)	13	7.4
Cannabis use, lifetime (Yes)	3	1.7
Cannabis use, past year (Yes)	2	1.1
Cannabis use, past 30 days (Yes)	2	1.1

Note. SD = standard deviation; SDQ = Strengths and Difficulties Questionnaire. Percentages computed on total sample (N = 177).

**Table 2 jcm-14-07177-t002:** Ordinary least-squares (OLS) predictors of SDQ TD in adolescents attending a pediatric clinic.

Predictor	B(SE)		95% CI	*p* _BH_	*β*	Δ ^a^
Intercept	34.24	(5.24)	[23.73, 44.21]	—	—	—
Age	−0.39	(0.34)	[−1.03, 0.33]	0.362	−0.08	0.28
Sex	2.30	(0.70)	[0.91, 3.70]	**0.002**	0.21	0.14
Chronic condition	−0.16	(0.68)	[−1.45, 1.21]	0.788	−0.02	0.33
Italian citizenship	0.43	(1.00)	[−1.77, 2.18]	0.788	0.02	0.17
Mother-relationship satisfaction	−0.68	(0.56)	[−1.61, 0.61]	0.362	−0.10	0.20
Father-relationship satisfaction	−1.00	(0.47)	[−1.87, −0.03]	0.054	−0.18	0.11
Friendship satisfaction	−2.05	(0.35)	[−2.71, −1.31]	**<0.001**	−0.33	0.04
Eating-disorder risk	1.39	(0.87)	[−0.35, 3.04]	0.210	0.11	0.06
Prosocial behavior score	−0.64	(0.20)	[−1.04,−0.27]	**0.006**	−0.23	0.07
Alcohol use (30 d)	0.45	(1.46)	[−2.24, 3.56]	0.788	0.02	0.01
Nicotine use (30 d)	2.29	(1.61)	[−0.77, 5.55]	0.298	0.11	0.73
Model fit: R^2^ = 0.41, adj. R^2^ = 0.37; *F*(11, 165) = 10.49, *p* < 0.001.						

Note. B(SE) = unstandardized coefficient with bootstrap SE; 95% CI = BCa pairs-bootstrap; *p*_BH_ = Benjamini–Hochberg-adjusted bootstrap *p*-value (two-sided), across 11 predictors; intercept excluded; *β* = standardized OLS coefficient; Δ = B_robust_ − B_OLS_; Sex (female): coded 1 = female, 0 = male; ᵃ Predictors with |Δ| ≤ 0.10 were deemed stable to outlier down-weighting. Significant *p* values are reported in bold.

## Data Availability

All the available data are included in the manuscript and in Appendix A.

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
