# Peer review of "Peer Relationships and Psychosocial Difficulties in Adolescents: Evidence from a Clinical Pediatric Sample"

_jcm, 2025, doi:10.3390/jcm14207177_

Round 1

Reviewer 1 Report

Comments and Suggestions for Authors

In order to improve the quality of the manuscript, certain aspects need to be revised.

  1. The research background section provides a comprehensive overview of the importance of adolescent mental health and its influencing factors, but could further emphasize the unique value of studying these factors within the clinical pediatric population.
  2. However, the research hypotheses could be more specific. For example, “H1: ” should be stated, along with the theoretical basis for each hypothesis.

3.The sample selection process is clearly described, but the sample size for this study may be insufficient.

  1. Consider adding a section on the previous applicability of the study's measurement tools within the Italian cultural context.
  2. The advantages of the statistical methods used in this study could be explained in Section 2.3 Statistical Analysis.
  3. Figure 1 could benefit from incorporating more visual elements.
  4. The Discussion section provides a comprehensive interpretation of the findings but requires more comparison with existing literature.
  5. Consider variations in friendship satisfaction across different cultural or clinical contexts.
  6. Propose more concrete intervention strategies, such as methods to enhance adolescent friendship quality in school or clinical settings.
  7. The study limitations section mentions constraints like the cross-sectional design but does not sufficiently discuss how these limitations impact the conclusions.

Author Response

In order to improve the quality of the manuscript, certain aspects need to be revised.
Re: Thank you for your comments. We revised our manuscript according to the suggestions.

1.    The research background section provides a comprehensive overview of the importance of adolescent mental health and its influencing factors, but could further emphasize the unique value of studying these factors within the clinical pediatric population.
Re: A paragraph has been added according to the suggestion (p. 3).

2.    However, the research hypotheses could be more specific. For example, “H1: ” should be stated, along with the theoretical basis for each hypothesis.
Re: The text has been revised according to the suggestion (p. 3).

3.The sample selection process is clearly described, but the sample size for this study may be insufficient.
Re: We discussed this issue in the study limitations (p. 11).

4.    Consider adding a section on the previous applicability of the study's measurement tools within the Italian cultural context.
Re: We appreciate this helpful suggestion. In response, we added two brief note in Section 2.2 to clarify that all measurement tools (SDQ and SCOFF) have been previously used in Italian adolescent samples within the Italian context (p. 4).

5.    The advantages of the statistical methods used in this study could be explained in Section 2.3 Statistical Analysis.
Re: We thank the reviewer for this useful comment. As suggested, we have added a brief paragraph at the end of Section 2.3 to clarify the rationale for our analytical strategy (p. 5). In particular, we now highlight how the combination of OLS and robust MM regression, together with bootstrap inference and FDR correction, was designed to strengthen the robustness and interpretability of results and to address common features of real-world clinical data, such as outliers, non-normality, and limited sample size.

6.    Figure 1 could benefit from incorporating more visual elements.
Re: A new Figure 1 has been prepared according to the reviewer’s suggestion (p. 9).

7.    The Discussion section provides a comprehensive interpretation of the findings but requires more comparison with existing literature.
Re: Revised as suggested (pp. 9-10).

8.    Consider variations in friendship satisfaction across different cultural or clinical contexts.
Re: Revised as suggested (p. 10).

9.    Propose more concrete intervention strategies, such as methods to enhance adolescent friendship quality in school or clinical settings.
Re: Revised as suggested (p. 11).

Reviewer 2 Report

Comments and Suggestions for Authors

The study provides important insights into adolescent mental health, particularly the role of friendship satisfaction in mitigating psychological difficulties. By addressing the suggested revisions, the manuscript could further strengthen its contribution to the field.

  1. There are some areas where the narrative could be more streamlined, especially in the introduction and methods sections, to ensure that the significance of the research is communicated more clearly.
  2. The flow between the literature review and the research hypothesis can be enhanced by more directly linking the previous studies to the objectives of your current research.
  3. The abstract effectively summarizes the study, though some of the statistical information (e.g., the percentage of variance explained) may be more appropriate in the results section.
  4. The introduction could benefit from a more detailed explanation of why the study population (clinical pediatric sample) is unique compared to other adolescent mental health studies, specifically focusing on the clinical aspect.
  5. The choice of single-item Likert-type ratings for parental and friendship satisfaction is a potential limitation, as these may lack the depth of multi-item scales. Consider discussing this limitation further in the manuscript.
  6. Further discussion of potential confounding variables, such as socioeconomic status, could add depth to the interpretation of the results.
  7. The cross-sectional nature of the research is a key limitation that could benefit from more detailed discussion, particularly in terms of causal inference.
  8. The low prevalence of substance use may reduce the power of the study in examining its impact on adolescent mental health. This should be highlighted more explicitly in the limitations.

Minor:

  1. There are minor typographical errors in some parts of the manuscript (e.g., "psychosocial well-being" should be "psychosocial wellbeing").
  2. Consider including more recent references in the literature review to provide a comprehensive view of the current research landscape.

Author Response

The study provides important insights into adolescent mental health, particularly the role of friendship satisfaction in mitigating psychological difficulties. By addressing the suggested revisions, the manuscript could further strengthen its contribution to the field.
Re: Thank you for your comments. We revised the manuscript according to your suggestions.

1.    There are some areas where the narrative could be more streamlined, especially in the introduction and methods sections, to ensure that the significance of the research is communicated more clearly.
Re: Introduction and Methods have been improved as recommended (pp. 3-5).

2.    The flow between the literature review and the research hypothesis can be enhanced by more directly linking the previous studies to the objectives of your current research.
Re: Revised as suggested (p. 3).

3.    The abstract effectively summarizes the study, though some of the statistical information (e.g., the percentage of variance explained) may be more appropriate in the results section.
Re: Abstract results have been rewritten (p. 1).

4.    The introduction could benefit from a more detailed explanation of why the study population (clinical pediatric sample) is unique compared to other adolescent mental health studies, specifically focusing on the clinical aspect.
Re: Clarified as recommended. 

5.    The choice of single-item Likert-type ratings for parental and friendship satisfaction is a potential limitation, as these may lack the depth of multi-item scales. Consider discussing this limitation further in the manuscript.
Re: Considered in study limitations (p. 12).

6.    Further discussion of potential confounding variables, such as socioeconomic status, could add depth to the interpretation of the results.
Re: Considered in study limitations (p. 12).

7.    The cross-sectional nature of the research is a key limitation that could benefit from more detailed discussion, particularly in terms of causal inference.
Re: Considered in study limitations (p. 12).

8.    The low prevalence of substance use may reduce the power of the study in examining its impact on adolescent mental health. This should be highlighted more explicitly in the limitations.
Re: Considered in study limitations (p. 12).

Minor:
1.    There are minor typographical errors in some parts of the manuscript (e.g., "psychosocial well-being" should be "psychosocial wellbeing").
Re: Thank you for your comment. The text has been revised by an English mothertongue.

2.    Consider including more recent references in the literature review to provide a comprehensive view of the current research landscape.
Re: A total of 19 new references have been added (pp. 13-18).

Reviewer 3 Report

Comments and Suggestions for Authors

Firstly, I congratulate the authors on their choice of subject matter, which focuses on the adolescent population.
Overall, the manuscript is very well presented. All chapters are well explained and detailed.
Ethical considerations are well explained.
Suggestions:
- The title could be revised, as mental health is not objectively assessed. Other indicators of mental health are assessed.
- The bibliography could be more recent, as only one reference is less than five years old. The research could be improved to find more up-to-date scientific evidence.

Author Response

Firstly, I congratulate the authors on their choice of subject matter, which focuses on the adolescent population.
Overall, the manuscript is very well presented. All chapters are well explained and detailed.
Ethical considerations are well explained.
Re: Thank you very much for your positive evaluation. We improved the text according to your suggestions.

Suggestions:
- The title could be revised, as mental health is not objectively assessed. Other indicators of mental health are assessed.
Re: The title has been revised according to your comment (p. 1).

- The bibliography could be more recent, as only one reference is less than five years old. The research could be improved to find more up-to-date scientific evidence.
Re: A total of 19 new references have been added (pp. 13-18).

Round 2

Reviewer 2 Report

Comments and Suggestions for Authors
  1. Low resolution of figure1.
  2. The introduction need to be simplified.
  3. The disscussion need tobe simplified.
  4. Is there any medication or drug administration in the adolescents?
  5. How is the BMI scores of the participants? The author should consider this factor.

Author Response

Re: Thank you for your suggestions. We revised the manuscript accordingly.

1.    Low resolution of figure1.
Re: Improved (p. 9).

2.    The introduction need to be simplified.
Re: Done as recommended (pp. 1-3).

3.    The disscussion need tobe simplified.
Re: Done as recommended (pp. 8-9).

4.    Is there any medication or drug administration in the adolescents?
Re: We did not collect information on these topics. It has been added among study limitations (p. 9).

5.    How is the BMI scores of the participants? The author should consider this factor.
Re: We did not collect information on BMI. It has been added among study limitations (p. 9).